Effect of quality sleep on basketball three-point shooting outcomes: the mediating role of athletic mental energy in a cross-sectional study of collegiate athletes

Chan Shu-Yueh 1
Shen Wei-Jiun shenweichjiun@gmail.com 1 2
Lo Shin-Liang 1
Hsieh Yun Che 1 3
Lu Frank J.H. 1
Kuan Garry 4
1 Department of Physical Education, Chinese Culture University , Taipei , Taiwan
2 Graduate Institute of Physical Education, National Taiwan Sport University , Taoyuan , Taiwan
3 Department of Sport Sciences, Army Academy, R.O.C , Taoyuan , Taiwan
4 Exercise and Sports Science Programme, School of Health Sciences, Universiti Sains Malaysia , Kubang Kerian , Kelantan , Malaysia
van den Hoek Daniel
Electronic publication date: 2025 Nov 19
Publication date: 2025
Volume: 13
Electronic Location ID: e20355
Received 2025 May 14; Accepted 2025 Oct 16
Copyright: ©2025 Chan et al.
Copyright year: 2025
Copyright holder: Chan et al.
License: This is an open access article distributed under the terms of the Creative Commons Attribution License, which permits unrestricted use, distribution, reproduction and adaptation in any medium and for any purpose provided that it is properly attributed. For attribution, the original author(s), title, publication source (PeerJ) and either DOI or URL of the article must be cited.
License URL: https://creativecommons.org/licenses/by/4.0/

Keywords: Quality of life, Sports excellence, Optimal functioning, Peak performance

Funding: The National Science and Technology Council, R.O.C. No. NSTC 113-2410-H-034-021- This work was supported by the National Science and Technology Council, R.O.C. (No. NSTC 113-2410-H-034-021-). The funders had no role in study design, data collection and analysis, decision to publish, or preparation of the manuscript.

==============================
Introduction

Quality sleep is crucial for optimal sports performance, yet the psychological mechanisms underpinning the sleep-sports performance relationship require further examination.

Methods

This cross-sectional study explored the effects of athletic mental energy (AME) and sleep quality on basketball three-point shooting outcomes with a particular emphasis on the mediating role of AME. One hundred and forty-five collegiate basketball athletes (71 males and 74 females; Mage = 19.62 ± 1.35) with highly trained levels were recruited to evaluate sleep quality, AME, and basketball three-point shooting performance and percentage. Data were collected through validated questionnaires and a standardized three-point shooting test, and analyzed using structural equation modeling (SEM) with bootstrapping.

Results

Sleep quality was positively associated with basketball three-point shooting performance (r = 0.22, p = 0.007) and shooting percentage (r = 0.22, p = 0.009). AME partially and fully mediated these associations (indirect effect = 0.18, p = 0.031, 95% bias-corrected CI [0.02–0.42] for shooting performance; indirect effect = 0.27, p = 0.019, 95% bias-corrected CI [0.04–0.60] for shooting percentage).

Conclusions

The findings contribute to the literature on the relationship between sleep and competition-relevant sports performance and suggest AME as one of the potential psychological mechanisms underlying these associations. These results highlight the importance of considering athletes’ sleep quality and AME in performance contexts, while further research is needed to strengthen and generalize these conclusions.

Introduction

Sleep is a fundamental biological necessity for maintaining physiological and psychological well-being (Cappuccio et al., 2010). High-quality sleep enhances cognitive performance (Zavecz et al., 2020), emotional regulation, and overall health (Apinis-Deshaies, Trempe & Tremblay, 2023; Scott et al., 2021), whereas insufficient sleep impairs metabolic and immune functioning, elevates psychological distress, and increases cardiovascular risk (McAlpine et al., 2022; Halson et al., 2022; Tobaldini et al., 2017). In athletic contexts, sleep provides restorative functions that sustain the cognitive and physical capacities required for elite-level competition (Charest & Grandner, 2020; Doherty et al., 2021; Halson, 2014). Student-athletes are particularly vulnerable given sport demands, travel, academic and financial pressures, and chronotype differences, making insufficient and irregular sleep common in this population (Wilson et al., 2025). Many studies focusing on student-athletes have demonstrated that insufficient sleep detrimentally affects reaction time, attention, decision-making, energy levels, emotional regulation, endurance, movement precision, and motor skills (Hamlin et al., 2021; Roberts, Teo & Warmington, 2019; Vitale et al., 2019; Walsh et al., 2021). Conversely, sufficient sleep consistently enhances athletic performance across contexts, including anaerobic power (Lim et al., 2021), CrossFit performance (Klier, Dörr & Schmidt, 2021), football outcomes (Brandt, Bevilacqua & Andrade, 2017), netball rankings (Juliff et al., 2018), and basketball shooting accuracy (Mah et al., 2011). Although some studies found no significant relationship between sleep and acute performance, suggesting sleep is one of multiple contributing factors (Fullagar et al., 2015; Lastella, Lovell & Sargent, 2014), meta-analyses affirm its overall importance. A synthesis of 45 studies confirmed that sleep deprivation significantly impairs speed, anaerobic endurance, maximal force, skill execution, and perceived exertion (Kong et al., 2025), while reviews further corroborate the detrimental effects of acute sleep restriction (Craven et al., 2022; Gong et al., 2024). Collectively, evidence highlights sleep as a fundamental determinant of both physical and cognitive dimensions of athletic performance.

Despite widespread recognition of the influence of sleep quality on sports performance, the psychological pathway linking sleep and sports performance is still unclear. One potential psychological construct, athletic mental energy (AME), may explain the sleep-sports performance relationship. According to Lu et al. (2018), the AME refers to “…an athlete’s perceived existing state of energy, which is characterized by its intensity in motivation, confidence, concentration, and mood.” And it comprises both cognitive factors (i.e., confidence, concentration, and motivation) and emotional factors (i.e., vigor, calm, and tirelessness) (Lu et al., 2018). Specifically, empirical studies support AME is associated with many psychological attributes, such as the adoption of adaptive dietary practices (Ilhan, 2020; Yildiz et al., 2020), imagery use (Kaplan & Bozdağ, 2022), courage, psychological skills (Islam, 2022; Islam, 2023), mindfulness, and flow states (Öner, 2022). Further, it has been found that AME predicts sports performance such as volleyball competition performance, martial arts medals, and track and field medals (Chiou et al., 2020; Chuang et al., 2022; Lu et al., 2018; Novan et al., 2023; Shieh et al., 2023). Although the direct relationship between AME and sleep has not been extensively investigated, several studies provide evidence regarding the association between sleep quality and components related to AME. Specifically, adequate or extended sleep has been positively associated with motivation (Mah et al., 2011), confidence (Tan et al., 2023), concentration (Fullagar et al., 2015), vigor (Lastella, Lovell & Sargent, 2014), and mood regulation (Charest & Grandner, 2020). In contrast, poor sleep quality and shorter sleep duration have been linked to adverse psychological outcomes, including increased anxiety (Fullagar et al., 2015), elevated stress and depression (Grandner et al., 2021), and greater fatigue and pain (Doherty et al., 2021). Collectively, these findings highlight the fundamental role of sleep in promoting psychological readiness and optimal functioning in sports, thereby suggesting a plausible psychological mechanism through which AME may operate.

In the realm of sports, mental energy is crucial for sustaining focus, regulating emotions, and enhancing performance. Insufficient sleep diminishes this energy, thereby impairing both cognitive and motor skills (Simpson, Gibbs & Matheson, 2017). According to the cognitive energetic theory, task engagement necessitates energy proportional to the task’s demands, with performance contingent upon the availability and allocation of energetic resources Sanders, 1983. From this standpoint, insufficient sleep diminishes the availability of these resources, thereby impairing cognitive functions such as attention, motivation, and emotional regulation (Kruglanski et al., 2012; Charest & Grandner, 2020). These resource-dependent processes conceptually overlap with AME, which encompasses athletes’ perceived levels of concentration, confidence, motivation, vigor, and calmness. Similarly, the strength model of self-control emphasizes that self-control is dependent on a limited mental resource, with depletion of this resource impairing operating process (Baumeister et al., 1998; Baumeister, Vohs & Tice, 2007; Muraven & Baumeister, 2000; Guarana et al., 2021) and athletic performance (Englert et al., 2015a; Englert et al., 2015b). A meta-analysis also found that prior exertion of self-control increases perceived effort and decreases self-efficacy, ultimately undermining performance (Hunte et al., 2021). Self-determination theory further emphasizes that intrinsic motivation and competence are essential for sustained engagement and optimal performance (Deci & Ryan, 2000), and evidence suggests that sleep quality replenishes these motivational resources (Tan et al., 2023). Likewise, conservation of resources theory (Hobfoll, 1989) indicates that individual strive to retain and restore valued resources, with sleep serving as a fundamental recovery mechanism. Collectively, sleep plays a critical role in restoring mental energy, psychological resources, motivation and self-control, which in turn enhances athletic performance (Balk & Englert, 2020; Englert, 2016; Bertrams, 2020). AME, a multifaceted psychological construct that consolidates these resource-based processes (Lu et al., 2018), therefore provides a theoretically grounded mediator between sleep and sports performance. From the standpoint of mental energy, resource, motivation, and self-control, the mediating role of constructs related to mental energy in the transition from sleep to work performance has been extensively documented (Henderson & Horan, 2021; Schmitt, Belschak & Den Hartog, 2017); however, this transition remains underexplored within the context of sports.

Furthermore, while existing research has reported significant associations between sleep quality, factors related to AME, and sport-specific outcomes (Fullagar et al., 2015; Lastella, Lovell & Sargent, 2014; Cunha et al., 2023), the predominant reliance on laboratory-based assessments, self-reports, and outcome-based evaluations such as anaerobic and fitness tests (Fullagar et al., 2015; Klier, Dörr & Schmidt, 2021; Lim et al., 2021), match results (Brandt, Bevilacqua & Andrade, 2017; Juliff et al., 2018), or self-assessed performance (Lastella, Lovell & Sargent, 2014; Mah et al., 2011) may limit ecological validity. Although such designs allow for experimental control, they may not fully replicate the dynamic, uncertain, and interactive nature of real competitive scenarios (Pinder et al., 2011; Araújo, Davids & Passos, 2007). As Glazier (2017) noted, behaviors elicited under standardized conditions do not always reflect authentic performance, underscoring the importance of tasks that approximate natural sport environments. In this regard, basketball three-point shooting has been proposed as a suitable metric, as it has been used to examine sleep effects (Mah et al., 2011; Miyaguchi et al., 2022), although limitations such as small samples, amateur participants, or simplified scoring methods constrain generalizability. From a motor behavior perspective, basketball three-point shooting can reflect coordination and consistency of motor patterns (Fan et al., 2024; Li et al., 2025) and may indicate perceptual-cognitive integration, encompassing technical execution, physical capacity, cognitive focus, and psychological readiness (Hughes & Bartlett, 2002). Analyses of elite competitions further suggest that basketball three-point performance has an important influence on game outcomes, representing both motor proficiency and decision-making under pressure (Sampaio et al., 2015). Methodologically, it may serve as a reliable index through comprehensive scoring that better captures variability and reflects both technical and cognitive demands (Lu et al., 2020). Therefore, basketball three-point shooting can be regarded not only as a measure of technical skill but also as a potential proxy for broader performance dimensions that are theoretically associated with sleep and AME.

In summary, although the importance of sleep quality for athletic performance is well recognized, the psychological pathways underlying this association remain underexplored. Existing evidence has established that sleep is positively associated with sport performance outcomes, that sleep is linked to energy-related psychological states such as vigor, motivation, and concentration, and that AME itself predicts athletic performance. AME therefore provides an integrative framework to examine how sleep translates into performance through multiple cognitive and emotional dimensions. Furthermore, basketball three-point shooting performance serves as an indicator of athletic skill and reflects varying degrees of motor control precision in real sport setting, thereby providing an appropriate framework for examining the mediating relationship among sleep, AME, and authentic sports performance. The novel contribution of the present study is to extend this prior work by explicitly testing AME as a mediator between sleep quality and basketball-specific performance, using an ecologically valid three-point shooting task. Accordingly, present study pursues two primary objectives: (a) to examine the relationships among sleep quality, AME, and basketball three-point shooting outcomes; and (b) to investigate the mediating role of AME in the relationship between sleep quality and sports performance. It was hypothesized that sleep quality is positively associated with both athletes’ AME and basketball three-point shooting outcomes, and that AME would mediates the relationship between sleep quality and basketball three-point shooting performance and percentage.

Materials & Methods

Sample size estimation

The required sample size was estimated in advance using G*Power (Faul et al., 2009). The analysis suggested that at least 68 participants would be needed to test two predictors with a statistical power of 0.80, an alpha level of 0.05 and a medium effect size (f2 = 0.15). in addition, Fritz & Mackinnon (2007) recommended a minimum of 71 participants to adequately conduct mediation analyses with a medium effect size of 0.26. To further justify our sample size, we also conducted a Monte Carlo simulation for mediation analysis using standardized parameters (i.e., Var(X) = Var(M) = Var(Y) = 1), with anticipated path coefficients set at a = 0.30 and b = 0.30, consistent with prior findings linking sleep-related factors to AME (Tan et al., 2023) and AME to sport performance (Lu et al., 2018; Chuang et al., 2022). With 10,000 replications and bias-corrected bootstrap confidence intervals at α = 0.05, the required sample size to achieve approximately 0.80 power was estimated to range from 66 to 75 participants. Therefore, the obtained sample of 68 participants was deemed adequate to detect the hypothesized indirect effects.

Participants

To mitigate the floor effect in the measurement of basketball shooting performance (Thomas et al., 2015), inexperienced and novice basketball players were excluded. A total of 154 collegiate basketball players (75 males and 79 females; M age = 19.64 ± 1.34) from 16 universities were recruited from Taiwan’s University Basketball Association (UBA), specifically Division I and Division II leagues. Based on the performance-level classification proposed by McKay et al. (2022), these athletes approximately corresponded to Tier 3 (comparable to NCAA Division II–III). Participants were required to be current collegiate basketball athletes who had engaged in formal team training for at least six months and who considered their physical condition sufficient to complete both the questionnaires and the basketball three-point shooting test. Exclusion criteria included incomplete responses to the survey instruments or failure to complete the shooting task. No additional restrictions were applied based on sleep variability, anthropometric characteristics, or injury status, as participation was determined by athletes’ self-assessment of their ability to complete the study procedures.

Procedures

Ethical approval for the current study was secured from Antai Medical Care Cooperation Antai Tian-Sheng Memorial Hospital Institutional Review Board prior to the study’s administration (approval number: TSMHIRB-23-090-B). The researchers contacted the basketball teams of various universities and colleges to obtain the consent of the coaches and players. During the initial meeting, the researchers introduced themselves to the players and explained the purpose of the study in detail. We reiterated that participation in the study was not mandatory and that participants could withdraw at any time during the study. Participants were informed that all data collected would be used solely for this study and would not be disclosed. Participants who wished to participate were asked to sign an informed consent form.

All testing was conducted during the pre-season training period, scheduled 1–2 h before each team’s regular training session, either in the morning or afternoon. After providing informed consent, participants first completed a demographic questionnaire, the Pittsburgh Sleep Quality Index (PSQI; Buysse et al., 1989), and the Athletic Mental Energy Scale (AMES; Lu et al., 2018). Following questionnaire administration, participants engaged in a 5-minute individual warm-up, attempting two practice shots from each of the five positions along the three-point line (a total of ten shots). Thirty seconds after completing the warm-up, the main shooting test was initiated. To enhance the ecological validity and approximate competitive conditions, the shooting procedure was conducted in small groups of three participants from the same team. Within each group, one participant served as the shooter, one as the passer, and one as the retriever, while a trained research assistant, who served as the designated scorekeeper, recorded both the graded scores and the shooting percentage of the shooter’s attempts. Ten basketballs were prepared and placed on a ball rack positioned midway between the free-throw line and the basket (male participants used Molten Size 7 basketballs, and female participants used Molten Size 6 basketballs). At the beginning of each trial, the shooter stood ready without a ball, received a bounce pass from the passer, and took a shot. Following each attempt, the retriever collected all balls, whether successful or missed, and returned them to the rack. The passer, positioned beside the rack, then delivered the subsequent bounce pass immediately after the previous shot had bounced on the floor. The shooter proceeded in a clockwise direction from position 1 to position 5, attempting five shots at each location (25 shots in total). The shooter was not required to retrieve rebounds, and each shot had to be released within two seconds of receiving the ball to simulate decision-making speed under game-like pressure. The shooting performance was evaluated using the standardized scoring system developed by Lu et al. (2020), which grades each shot from 0 to 5 based on precision (e.g., “swish,” rim in, backboard, miss). Two video devices were also placed at the intersections of the painted area and the baseline to verify scoring accuracy, without capturing identifiable images of the players. After each shooter completed the test, the roles rotated: the shooter became the retriever, the retriever became the passer, and the passer became the shooter. Before each new shooter began their test, they performed an individual warm-up using the same 5-minute routine. After all three participants in the group completed their tests, the next group was called, and each testing section lasted approximately 30 min. The configuration of the court, personnel roles, shooting positions, and equipment setup are illustrated in Fig. 1.

Figure 1 Basketball court, shooting positions, personnel role and equipment setup.

The numerical order within the black circles represents the sequence of shooting attempts.

Apparatus

This study was conducted on the indoor basketball courts of various universities and colleges. The specifications for the basketball, backboard dimensions, hoop height, and three-point line distance all conformed to the standards of the International Basketball Federation (FIBA). A detailed description of each component is provided below:

1. Basketball: Following FIBA regulations, male participants used the Molten Size 7 basketball, while female participants used the Molten Size 6 basketball.

2. Backboard: The backboard was made of transparent acrylic with a thickness of three cm. It measured 1.80 m in width and 1.05 m in height. The lower edge of the backboard was positioned 2.90 m above the ground.

3. Basketball hoop: The rim had an inner diameter of 45 cm, with its upper edge 3.05 m above the ground. The distance between the backboard and the inner edge of the rim at its nearest point was 15 cm.

4. The three-point line: The three-point line was drawn by forming a semicircle with a 6.75 m radius from the rim’s center, on the side facing away from the backboard. As it approached the sidelines, the curve transitioned into a straight line that runs parallel to the sidelines at a distance of 0.75 m from them.

5. Shooting locations: Five shooting spots were placed along the three-point line at angles of 0°, 45°, 90°, 135°, and 180°, as illustrated in Fig. 1.

Measurement

Demographic questionnaire: The demographic questionnaire was designed to collect participants’ sex, age, basketball position in basketball, level of division, and years of athletic experience, including training hours per session and days per week.

Sleep Quality: The PSQI (Buysse et al., 1989) is a self-administered questionnaire comprising 18 items evaluating overall sleep quality over the preceding month. The PSQI includes seven components: (1) subjective sleep quality, (2) sleep latency, (3) sleep duration, (4) sleep efficiency, (5) sleep disturbances, (6) use of sleep medication, and (7) daytime dysfunction. Each component of the PSQI is rated on a 4-point Likert scale, with scores ranging from 0 to 3. Traditionally, the PSQI employs a reverse scoring system, whereby higher scores indicate poorer sleep quality. In the present study, we adhered to this reverse scoring methodology. The cumulative scores yield a global PSQI score, which ranges from 0 to 21, with higher scores indicating superior sleep quality. The total PSQI score was utilized for the primary analysis, except for the variable concerning sleep medication, which exhibited a extreme skewness of −10.63 (only two participants rated 1). The Cronbach’s alpha for the current sample was 0.70.

AME: The AMES (Lu et al., 2018) was administered to assess participants’ AME. The AMES includes three items in each of six factors: vigor (e.g., “Either in competition or training, I feel full of energy”), motivation (e.g., “I am full of passion to attend my sports”), confidence (e.g., “I can control all sports movements and skills”), concentration (e.g., “There is nothing distracting me in competition”), tireless (e.g., “No matter how long the training lasts, I don’t feel tired”) and calm (e.g., “Facing upcoming competitions, I don’t feel anxious.”). Participants were introduced to identify “how do you feel right now in sports training/competition” and respond to each item. The AMES uses a six-point Likert scale to rate participants’ responses, with “1” representing “not at all,” and “6” representing “completely so.” A higher score represents higher AME. The total AMES score was used for the primary analysis. Cronbach’s αs in the present sample were 0.87 for vigor, 0.80 for motivation, 0.77 for confidence, 0.85 for concentration, 0.88 for tireless, 0.87 for calm, and 0.94 for total AMES.

Basketball three-point shooting outcomes: The basketball three-point scoring system (Lu et al., 2020) was applied to measure participants’ three-point shooting performance. The criteria are as follows: 5 points: A “swish” shot, where the ball directly enters the basket without contacting either the rim or the backboard. Four points: A “rim in” shot, where the ball first contacts the rim before scoring. Three points: A “back shot”, where the ball first contacts the backboard before scoring. Two points: A “rim out” shot, where the ball first contacts the rim but does not score. One point: The ball first contacts the backboard but does not score. Zero points: A missed shot where the ball in which the ball neither contacts the rim nor the backboard. The score of each position ranged from 0 to 25 points and the total score ranged from 0 to 125 points. The total score was applied for the major analysis. Additionally, we recorded the total basketball three-point shooting percentage, calculated as the number of successful shots divided by the total number of attempts, multiplied by 100%, to provide a more intuitive measure of shooting outcome.

Statistical analyses

Prior to conducting the primary analyses, data screening was conducted using Little’s test for missingness, Z-values for univariate outliers, Mahalanobis distance for multivariate outliers, and the Kolmogorov–Smirnov test for normality (Tabachnick & Fidell, 2018). Path analysis in structural equation modeling (SEM) with bootstrapping was performed to test the mediation (Hayes, 2009; Nevitt & Hancock, 2001; Preacher & Hayes, 2008). Considering that the main variables might differ across sex, competitive level and playing position, and may be associated with athletic experiences (Clemente et al., 2019; Wilson et al., 2025; Xu & Li, 2024; Zhai et al., 2021), and that these factors could potentially influence the mediation, t-test, ANOVA and correlation analysis were conducted to examine their main effects. Model 7 and Model 14 in PROCESS (Hayes, 2022) were applied to further test the interaction effects after the mediation analysis. The statistical significance level was set at .05, and both p-values and 95% bias-corrected confidence intervals (CI) were reported to evaluate the direct and indirect mediation effects (Preacher & Hayes, 2008). In addition, the proportion mediated (PM) value (defined as the ratio of the indirect effect to the total effect) was calculated as an effect size indicator of mediation. According to prior guidelines, PM values of approximately 0.20, 0.50, and 0.80 can be interpreted as small, medium, and large mediation effects, respectively (Preacher & Kelley, 2011). All analyses were conducted using IBM SPSS Statistics for Windows (Version 29.0) and AMOS (Version 29.0).

Results

Data screening

Following the data screening process, Little’s test indicated 63 instances of random missingness across five cases, χ2(35) = 55.84, p = 0.371. Additionally, three univariate outliers (with absolute Z-scores exceeding 2.58) and two multivariate outliers (identified by Mahalanobis distance greater than 24.02, all p < 0.05) were excluded from subsequent analyses. Ultimately, nine cases were excluded, resulting in a final sample of 145 participants (71 males and 74 females; Mage = 19.62 ± 1.35). On average, they had 7.51 ± 2.75 years of experience in basketball and trained approximately 3.81 h per day (SD = 1.30). All primary variables, except for the use of sleep medication, exhibited a lightly non-normal distribution, with skewness values ranging from −0.81 to 1.50, as confirmed by the Kolmogorov–Smirnov test (all p < 0.01). The use of sleep medication was excluded from analysis due to its severe skewness (M = 2.97, SD = 0.26, skewness = −10.63). Additionally, to address non-normal distribution in model estimation, path analysis in SEM with Bollen-Stine bootstrapping (n = 5,000) was performed (Bollen & Stine, 1992; Nevitt & Hancock, 2001).

Descriptive statistics and correlations

The final sample included 145 participants (49% male and 51% female) with a Mage of 19.62 years (SD = 1.35). Participants’ average height and weight were 174.61 cm (SD = 10.97) and 68.97 kg (SD = 12.44), respectively (MBMI = 22.46, SD = 2.19). Participants reported approximately 7.51 years (SD = 2.75) of basketball training experience, with a training load of 3.81 h per day (SD = 1.30) and 5.19 sessions per week (SD = 2.24). Of these athletes, 57% competed in UBA Division I and 43% in Division II, while 43% were guards, 40% were forwards, and 17% were centers. For the primary variables, the PSQI score averaged 11.65 (SD = 14.35), the AMES score averaged 68.78 (SD = 14.35), basketball three-point shooting performance averaged 80.14 (ranged from 44 to 102, SD = 11.48), and shooting percentage averaged 47.97% (ranged from 8% to 84%, SD = 15.95). Detailed demographic and descriptive characteristics are presented in Tables 1 and 2.

Table 1 The frequency distribution, mean and standard deviation of athlete’s demographics (N = 145).

Characteristics	n (%)	M (SD)	
Sex			
Male	71 (48.97%)	–	
Female	74 (51.03%)	–	
Age	–	19.62 (1.35)	
Height	–	174.61 (10.97)	
Weight	–	68.97 (12.44)	
BMI		22.46 (2.19)	
Competitive level			
Division 1	83 (57.24%)	–	
Division 2	62 (42.76%)	–	
position			
Guard	62 (42.76%)	–	
Forward	58 (40.00%)	–	
Center	25 (17.24%)	–	
Training year	–	7.51 (12.44)	
Training hour per day		3.81 (1.30)	
<2 h	32 (22.07%)	–	
2–4 h	67 (46.21%)	–	
>4 h	46 (31.72%)	–	
Training times per week		5.19 (2.24)	
<4 times	49 (33.79%)	–	
5–6 times	73 (50.34%)	–	
>7 times	23 (15.86%)	–	

Table 2 The mean and standard deviation of major variables, and bivariate correlations among PSQI, AMES, three-point shooting performance and percentage.

	1	2	3	4	5	6	7	8	9	10	11	12	13	14	15	16	
Sleep quality (1)	–																
Sleep latency (2)	0.60***	–															
Sleep duration (3)	0.29***	0.31***	–														
Sleep efficiency (4)	0.16	0.34***	0.60***	–													
Sleep disturbance (5)	0.33***	0.26**	0.12	0.08	–												
Daytime dysfunction (6)	0.49***	0.23**	0.29**	−0.01	0.25**	–											
Vigor (7)	0.24**	0.30***	0.18*	0.14	0.11	0.35***	0.87										
Confidence (8)	0.30***	0.29***	0.11	0.09	0.16	0.28***	0.66***	0.77									
Motivation (9)	0.18*	0.24**	0.01	−0.03	0.09	0.16	0.69***	0.68***	0.80								
Concentration (10)	0.29***	0.30***	0.10	0.03	0.21*	0.29***	0.62***	0.68***	0.63***	0.85							
Tireless (11)	0.15	0.31***	0.03	0.04	0.18*	0.23**	0.64***	0.57***	0.46***	0.57***	0.88						
Calm (12)	0.16	0.17*	0.02	0.05	0.15	0.22**	0.63***	0.67***	0.56***	0.58***	0.51***	0.87					
PSQI (13)	0.73***	0.74***	0.71***	0.62***	0.45***	0.55***	0.34***	0.32***	0.17*	0.31***	0.24**	0.19*	0.70				
AMES (14)	0.26**	0.39***	0.09	0.06	0.19*	0.31***	0.85***	0.86***	0.81***	0.84***	0.76***	0.81***	0.32***	0.94			
Three-point shooting performance (15)	0.16	0.25**	0.13	0.14	0.02	0.11	0.23**	0.20*	0.26**	0.19*	0.03	0.13	0.22**	0.21*	–		
Three-point shooting percentage (16)	0.14	0.22**	0.12	0.13	0.03	0.12	0.25**	0.22**	0.26**	0.21*	0.20	0.13	0.22**	0.22**	0.97***	–	
M	1.79	1.72	1.95	2.43	1.86	1.90	11.51	11.93	13.70	11.61	8.32	11.72	11.65	68.78	80.14	47.97	
SD	0.77	0.97	0.93	0.90	0.55	0.76	2.67	2.55	2.82	3.32	3.05	3.10	14.35	14.35	11.48	15.95	
Notes.

Cronbach’s α of each measure are presented in bold on the diagonal.

PSQI total score of Pittsburgh Sleep Quality Index except “use of sleep medication”

AMES Athletic Mental Energy Scale

* p < 0.05.

** p < 0.01.

*** p < 0.001.

The correlation analysis revealed that both the PSQI score and the AMES were positively correlated with basketball three-point shooting performance and shooting percentage (r = 0.21–0.22, p = 0.001–0.013). In terms of specific dimensions, sleep quality was significantly correlated with vigor (r = 0.24, p = 0.004), confidence (r = 0.30, p < 0.001), motivation (r = 0.18, p = 0.030), and concentration (r = 0.29, p < 0.001). Sleep latency was positively correlated with vigor (r = 0.30, p < .001), confidence (r = 0.29, p < 0.001), motivation (r = 0.24, p = 0.002), concentration (r = 0.30, p < .001), tireless (r = 0.31, p < 0.001), calm (r = 0.17, p = 0.038), basketball three-point shooting performance (r = 0.25, p = 0.002) and shooting percentage (r = 0.22, p = 0.009). Sleep duration was positively correlated with vigor (r = 0.18, p = 0.036), whereas sleep disturbance were positively correlated with concentration (r = 0.21, p = 0.011) and tireless (r = 0.18, p = 0.033). Daytime dysfunction showed positive correlations with vigor (r = 0.35, p < 0.001), confidence (r = 0.28, p < 0.001), concentration (r = 0.29, p < 0.001), tireless (r = 0.23, p = 0.005), and calm (r = 0.22, p = 0.009). Furthermore, vigor (r = 0.23, p = 0.005), confidence (r = 0.20, p = 0.015), motivation (r = 0.26, p = 0.001), and concentration (r = 0.19, p = 0.023) were positively correlated with basketball three-point performance. Similarly, basketball three-point shooting percentage was positively associated with vigor (r = 0.25, p = 0.003), confidence (r = 0.22, p = 0.007), motivation (r = 0.26, p = 0.002), and concentration (r = 0.21, p = 0.011). The means and standard deviations for the primary variables, along with their zero-order correlations, are presented in Table 2.

Direct and indirect effects of the mediation model

The path analysis in SEM with Bollen-Stine bootstrapping revealed partial mediation among perceived sleep quality, AME, and basketball three-point shooting performance, and complete mediation among perceived sleep quality, AME, and basketball three-point shooting percentage (Fig. 2). The direct effect of perceived sleep quality on basketball three-point performance was 0.64 (p = 0.035, 95% bias-corrected CI [0.05–1.23]), and the indirect effect of perceived sleep quality on basketball three-point performance through AME was 0.17 (p = 0.032, 95% bias-corrected CI [0.02–0.41], PM value = 0.21). Regarding basketball three-point shooting percentage, the direct effect of perceived sleep quality was 0.77 (p = 0.063, 95% bias-corrected CI [−0.05, 1.59]), and the indirect effect through AME was 0.27 (p = 0.019, 95% bias-corrected CI [0.04–0.60], PM value = 0.26). Specifically, the association between sleep quality and three-point shooting performance was mediated by vigor (estimate = 0.22, p = 0.019, 95% bias-corrected CI [0.04–0.49], PM value = 0.27), confidence (estimate = 0.17, p = 0.037, 95% bias-corrected CI [0.01–0.40], PM value = 0.21), and motivation (estimate = 0.14, p = 0.032, 95% bias-corrected CI [0.01–0.39], PM value = 0.17), whereas the association between sleep quality and basketball three-point shooting percentage was mediated by vigor (estimate = 0.35, p = 0.010, 95% bias-corrected CI [0.08–0.73], PM value = 0.33), confidence (estimate = 0.28, p = 0.015, 95% bias-corrected CI [0.05–0.62], PM value = 0.27), motivation (estimate = 0.20, p = 0.031, 95% bias-corrected CI [0.13–0.52], PM value = 0.19) and concentration (estimate = 0.25, p = 0.029, 95% bias-corrected CI [0.02–0.60], PM value = 0.24), while tireless and calm did not reach significance for either outcome. The perceived sleep quality’s unstandardized total, direct and indirect effects on three-point performance and shooting percentage are presented in Table 3.

Figure 2 The mediations among sleep quality, athletic mental energy and three-point shooting outcomes.

The upper panel illustrates the partial mediation model among PSQI, AMES, and three-point shooting performance, while the lower panel illustrates the complete mediation model among PSQI, AMES, and three-point shooting percentage. PSQI, total score of Pittsburgh Sleep Quality Index except “use of sleep medication”; AMES, Athletic Mental Energy Scale. *p < 0.05; ***p < 0.001.

Table 3 The direct and indirect effects with bootstrapped estimation of the mediation among PSQI, AMES, and 3-point performance.

Dependent variable	Mediator	Effects	Estimate	p	95% Bias-corrected CI	PM value	
					Lower	Upper		
Three-point shooting performance	Total score of AMES	Total	0.82	0.007	0.24	1.38	0.21	
Direct	0.64	0.035	0.04	1.23	
Indirect	0.18	0.031	0.02	0.42	
Vigor	Indirect	0.22	0.019	0.04	0.49	0.27	
Confidence	Indirect	0.17	0.037	0.01	0.40	0.21	
Motivation	Indirect	0.14	0.032	0.01	0.39	0.17	
Concentration	Indirect	0.15	0.074	−0.01	0.39	0.18	
Tireless	Indirect	0.02	0.744	−0.18	0.14	0.02	
Calm	Indirect	0.06	0.134	−0.02	0.22	0.07	
Three-point shooting percentage	Total score of AMES	Total	1.04	<0.001	0.26	1.79	0.26	
Direct	0.77	0.063	−0.05	1.59		
Indirect	0.27	0.019	0.04	0.60	
Vigor	Indirect	0.35	0.010	0.08	0.73	0.33	
Confidence	Indirect	0.28	0.015	0.05	0.62	0.27	
Motivation	Indirect	0.20	0.031	0.13	0.52	0.19	
Concentration	Indirect	0.25	0.029	0.02	0.60	0.24	
Tireless	Indirect	0.04	0.651	−0.27	0.17	0.04	
Calm	Indirect	0.09	0.103	−0.02	0.33	0.09	
Notes.

AMES Athletic Mental Energy Scale

PM proportion mediated

Italicized variables in the mediators column denote significant indirect effects.

Main effect of demographics variables primary outcomes

The independent t-test revealed that female players exhibited significantly shorter sleep latency (p = 0.046) and greater sleep disturbance (p = 0.018) than male counterparts. In contrast, male participants demonstrated higher levels of vigor, confidence, motivation, concentration, tireless, calm, shooting performance and percentage than female players, with all comparisons yielding p-values less than 0.001. Additionally, results from independent samples t-tests indicated no significant differences between Division 1 and Division 2 participants across all primary variables (all p > 0.05). The result of one way ANOVA revealed that guards and forwards outperformed centers in shooting performance, shooting percentage, vigor, motivation, and tireless (p = 0.005–0.015). Guards also reported higher concentration, confidence, and calm than centers (p = 0.010–0.045). In addition, correlation analyses showed that training year was positively associated with vigor (r = 0.19, p = 0.026), motivation (r = 0.23, p = 0.006), concentration (r = 0.17, p = 0.048), and calm (r = 0.26, p = 0.002). Training hours per session were positively correlated with shooting performance (r = 0.22, p = 0.009), shooting percentage (r = 0.26, p = 0.002), vigor (r = 0.20, p = 0.016), confidence (r = 0.19, p = 0.021), and motivation (r = 0.22, p = 0.007). Finally, training days per week were positively associated with shooting performance (r = 0.25, p = 0.003) and shooting percentage (r = 0.27, p = 0.001).

Interacting effects of demographic factors, sleep, and AME on primary variables

Potential moderating effects of sex, position, training years and training loads (training hours per session and days per week) were examined given their associations with the primary variables. Results from PROCESS (Hayes, 2022) indicated that neither sex (Model 7: F (1, 141) = 0.49, p = 0.487; Model 14: F (1, 140) = 0.30, p = 0.584 for three-point shooting performance; F (1, 140) = 0.17, p = 0.685 for basketball shooting percentage) nor position (Model 7: F (1, 141) = 0.47, p = 0.493; Model 14: F (1, 140) = 0.01, p = 0.914 for basketball three-point shooting performance; F (1, 140) < 0.01, p = 0.983 for three-point shooting percentage) significantly moderated the mediation. Similarly, results from Model 7 and Model 14 showed that training years, training hours per session, and training days per week did not moderate the relationships between sleep and shooting outcomes (p = 0.434–0.962). Therefore, all participants were retained in the primary analysis without stratification by sex, position, training years or training loads.

Discussion

This research investigated the mediating role of AME in the relationship between sleep quality and basketball three-point shooting performance and shooting percentage among collegiate basketball players. Employing structural equation modeling with bootstrapping techniques, the results revealed direct and indirect pathways mediated by AME. These findings suggest that sleep quality is associated with basketball three-point shooting outcomes and may indirectly contribute to performance through higher AME levels (PM value = 0.21 for shooting performance; PM value = 0.27 for shooting percentage). Notably, the average PSQI score in our sample indicated poorer sleep quality compared with previous samples of collegiate athletes, which is consistent with prior findings that more than 40% of collegiate athletes exceed the PSQI clinical cutoff for poor sleep quality (Mah et al., 2018; Walsh et al., 2021). Moreover, the mean item scores of the AMES in our sample (ranging from 3 to 4 on a 6-point scale) are consistent with those reported in prior studies using both the 18-item version with collegiate and disabled athletes (Lu et al., 2018; Chuang et al., 2022) and the 10-item version (AMES-10) with student-athletes (Shen et al., 2025). This convergence indicates that the observed values fall within expected ranges and strengthens the representativeness of the present sample.

Theoretical implications and contributions

In alignment with existing literature, our findings support a positive correlation between sleep quality and athletic performance (Brandt, Bevilacqua & Andrade, 2017; Juliff et al., 2018; Klier, Dörr & Schmidt, 2021; Lim et al., 2021; Mah et al., 2011). Prior studies, such as those conducted by Fullagar et al. (2015) and Charest & Grandner (2020), have established that sufficient sleep enhances cognitive functions vital for sports performance, including attention, reaction time, and decision-making. Our research corroborates the findings of Mah et al. (2011), who demonstrated that sleep duration was associated with enhanced basketball shooting accuracy, paralleling the associations observed in the basketball context. Additionally, Lim et al. (2021) emphasized the associations between sleep and anaerobic performance and technical skill execution, which are consistent with our conclusions that sleep-related mental energy is positively linked to basketball shooting outcomes. The findings of this study are consistent with research on AME (Chiou et al., 2020; Chuang et al., 2022; Lu et al., 2018; Novan et al., 2023; Shieh et al., 2023), the cognitive energetic model (Kruglanski et al., 2012), and the strength model of self-control (Baumeister et al., 1998; Baumeister, Vohs & Tice, 2007). These findings, along with the aforementioned theoretical frameworks, suggest that variables associated with mental energy, including psychological resources and self-regulatory capacities, are positively associated with athletic performance (Englert, 2016). Furthermore, current study also provides preliminary evidence for the mediating effect of AME, a multifaceted psychological construct that encompasses vigor, tireless, confidence, motivation, concentration, and calm. While previous research has documented isolated benefits of sleep on psychological states, such as mood enhancement, reduced anxiety, and improved cognitive functioning (Hamlin et al., 2021; Halson et al., 2022), our results offer a tentative understanding of how sleep-induced psychological states translate into measurable performance outcomes.

Moreover, recent studies underscore AME as a pivotal factor mediating various psychological and performance-related variables in sports contexts (Islam, 2022; Singh, Kaur Arora & Boruah, 2024). Our research builds upon these findings by demonstrating that sleep-related differences in AME are correlated with quantifiable performance metrics, including both basketball three-point shooting performance and percentage. Although the observed mediation effects were modest in size (PM value = 0.21 for shooting performance; PM value = 0.26 for shooting percentage), the consistency across both indices suggests a underlying mechanism. According to guidelines for mediation effect sizes (e.g., Preacher & Kelley, 2011), these values fall in the lower range, reinforcing the need for caution in interpreting the mediating role of AME. Although the observed mediation effects were modest in size, one possible explanation relates to the measurement approach. The AMES, while validated and widely used, is a broad composite measure of mental energy and may be less sensitive to specific processes such as attentional focus. In contrast, objective techniques such as electroencephalography (EEG) or eye-tracking have been shown to capture neural and behavioral markers of concentration and visual attention during sport tasks (Baumeister et al., 2008; Vine, Moore & Wilson, 2014). This methodological difference may partially explain the relatively small mediation effects observed in the present study. This finding should be interpreted cautiously; nonetheless, it supports the view that AME could function as a psychological pathway through which restorative sleep translates into sport-specific technical outcomes. Specifically, vigor, motivation, and confidence mediated the associations between sleep quality and both basketball three-point performance and shooting percentage, which can tentatively be understood from the perspectives of restorative sleep and self-efficacy theory. Restorative sleep replenishes physiological and psychological resources that support energetic activation (vigor) and sustained goal engagement (motivation) (Fullagar et al., 2015; Mah et al., 2011). These states, together with confidence, are central to self-efficacy processes, which facilitate greater persistence, attentional control, and perceived readiness for performance (Bandura, 1997; Moritz et al., 2000). From a self-determination perspective (Deci & Ryan, 2000), restorative sleep may also replenish intrinsic motivation and perceived competence, thereby sustaining engagement in demanding sport tasks. Likewise, conservation of resources theory (Hobfoll, 1989) emphasizes recovery and protection of valued resources, further supporting the role of sleep as a foundation for maintaining psychological energy in performance contexts. Hence, the presence of these mediators suggests, but does not confirm, that sleep quality may enhance performance by strengthening physically-energetic and efficacy-related dimensions of AME. In contrast, tireless showed no significant effect, which may be attributable to its conceptual overlap with vigor (Lu et al., 2018; Shen et al., 2025), making its unique contribution less discernible when vigor was included in the model. Calm also did not mediate the sleep-performance association. This dimension reflects an athlete’s composure in competitive or adversarial settings, whereas the present basketball three-point shooting task was designed as a skill test under relatively controlled conditions, potentially reducing the relevance of calmness in this context. Finally, concentration mediated only the association between sleep quality and basketball three-point shooting percentage but not shooting performance. This distinction may arise because the detailed scoring system of three-point performance (Lu et al., 2020) captures a broader range of motor control and stability, while shooting percentage provides a simpler accuracy index that is more directly influenced by attentional focus. These findings indicate that although sleep quality was positively related to all six AME dimensions, only certain dimensions functioned as mediators in the pathway from sleep to performance outcomes. In general, our findings align with prior work (Roberts, Teo & Warmington, 2019; Scott et al., 2021) highlighting the broader cognitive and psychological correlates of quality sleep, including psychological resilience, cognitive efficiency, emotional regulation and sports performance.

It should be noted that, compared to a basketball-specific sleep study by Mah et al. (2011), in which collegiate players increased their free-throw success from 7.9 to 8.8 out of 10 (approximately a 9% improvement) and their three-point field goal accuracy from 10.2 to 11.6 out of 15 (about a 9.2% increase), the strengths of the associations between sleep quality and our two basketball three-point shooting indicators were relatively modest. Consistent with this observation, the PM values of 0.21 (for shooting performance) and .26 (for shooting percentage) suggest small-to-moderate mediation effects of AME. This finding highlights the importance of cautious interpretation: under conditions where the direct association between sleep quality and athletic performance is not as strong as in prior intervention-based studies, the mediating role of AME appears to be limited in magnitude rather than decisive. Similar modest, yet significant, effects have been observed in other empirical studies, where sleep extension or improved sleep hygiene yielded performance benefits with effect sizes ranging from trivial to medium depending on the outcome measure (e.g., Fullagar et al., 2015; Lastella, Lovell & Sargent, 2014; Roberts, Teo & Warmington, 2019). Taken together, these comparisons reinforce the need for a cautious interpretation: although AME contributes as a psychological pathway linking sleep and performance, its explanatory power may be constrained, and additional mechanisms beyond AME likely play a role in the sleep-performance relationship.

This study extends prior research by focusing on a sport-specific task performed in authentic competitive training settings. Much of the existing work on sleep and sport performance has relied on laboratory-based motor tasks or self-reports, which, although convenient, do not fully capture the cognitive, emotional, and motor complexities of actual sport environments (Pinder et al., 2011; Glazier, 2017). In line with prior basketball research, we examined shooting percentage as a supplementary outcome, given that it represents an intuitive and holistic indicator of performance frequently adopted in applied and game-analytic studies (Sampaio et al., 2015). Beyond this straightforward indicator, we also employed a detailed basketball three-point performance scoring system, following the approach of Lu et al. (2020), which allows for a more graded evaluation that captures the multifaceted aspects of motor control, perceptual-cognitive processing, and psychological readiness under pressure. Our findings indicated that sleep quality and AME are jointly associated with these interconnected elements, providing preliminary insights into the psychological mechanisms through which sleep enhances complex sport tasks. Taken together, these results indicate that AME could be regarded as one of several plausible pathways linking sleep and performance, but not as a decisive mechanism, underscoring the need for future research to examine additional mediating processes that may better account for the sleep-performance relationship.

Practical implications

The importance of sleep for athletic performance has been well documented, with evidence linking adequate sleep to improved recovery, motor learning, and cognitive functions essential for sport (Fullagar et al., 2015; Mah et al., 2011; Walsh et al., 2021). Our findings support this evidence by showing that sleep quality is related to basketball-specific performance, although the effect size was modest. Basic practices such as maintaining consistent sleep schedules and optimizing sleep environments remain important (Vitale et al., 2019; Cunha et al., 2023). Technology-based tools such as wearables and mobile applications can also assist by tracking sleep duration and variability (Charest & Grandner, 2020). However, excessive reliance on monitoring may increase anxiety and lead to orthosomnia, where preoccupation with sleep undermines actual rest (Trabelsi et al., 2023). Thus, while monitoring can be useful, it should be applied judiciously to maximize benefits and avoid drawbacks.

Beyond general sleep hygiene, this study highlights the practical importance of AME as a more proximal mechanism through which sleep may influence sport-specific performance. Sleep can be considered an upstream recovery factor that replenishes AME, and in turn, AME represents a comprehensive state that includes motivation, confidence, concentration, vigor, calmness, and tireless. Previous research has shown that higher levels of AME are associated with resilience to stress and more consistent performance under pressure (Shieh et al., 2023; Chuang et al., 2022; Singh, Kaur Arora & Boruah, 2024). Monitoring AME using validated scales (Lu et al., 2018; Shen et al., 2025; Wu et al., 2024) can therefore provide practitioners with actionable information on athletes’ readiness. In applied practice, strategies to strengthen AME dimensions include fostering intrinsic motivation through autonomy-supportive coaching and goal-setting (Ryan & Deci, 2020), enhancing confidence with mastery experiences and self-talk training (Hatzigeorgiadis et al., 2011), improving concentration through quiet-eye attention programs (Vine, Moore & Wilson, 2014), sustaining vigor through recovery protocols that combine sleep extension with balanced nutrition (Snijders et al., 2019), promoting calm with mindfulness and paced breathing to regulate pre-competition anxiety (Bühlmayer et al., 2017), and reducing perceptions of fatigue by adopting sleep banking and mindfulness-based techniques (Lastella et al., 2015; Coimbra et al., 2021). Taken together, these approaches indicate that the practical value of sleep may not only lie in direct physiological recovery but also in its capacity to enhance AME, which functions as a proximal pathway to athletic performance.

Limitations and suggestions for future research

Several limitations of the present study warrant careful consideration. First, the cross-sectional design and the use of a specific sample consisting of Taiwanese collegiate basketball athletes limit causal inference and reduce the generalizability of the findings. To address these limitations, future research should employ longitudinal or experimental designs and recruit more diverse populations across various cultural contexts to enhance causal inference and broader applicability. Second, the exclusive focus on basketball three-point shooting performance may not fully capture the multifaceted nature of sports performance, as it reflects coordination, motor consistency and perceptual-cognitive skills but does not represent explosive or contact-based action such as rebounding or physical positioning. Future studies could therefore incorporate a broader range of performance assessments. Third, there is a temporal mismatch among some variables. Specifically, the PSQI assesses retrospective sleep quality over the preceding month, whereas the AMES evaluates perceived AME over the most recent days, including the day of testing, thereby aligning more closely with the timing of the performance task. In contrast, the basketball three-point shooting test reflects performance on a single day, raising concerns about whether the monthly scope of the PSQI adequately represents participants’ immediate sleep status before testing. Future studies could therefore consider employing sleep measures that capture both chronic and acute dimensions, such as device-based or self-reported sleep duration from the night before testing, to achieve improved temporal alignment with short-term performance outcomes. Fourth, detailed information regarding participants’ injury history and general sport performance characteristics (e.g., competitive results, physical skills) was not collected, which may be associated with psychological and performance outcomes. Future research should include such measures to provide a more comprehensive characterization of athlete samples. Fifth, both the PSQI and the AMES relied on self-reported assessments, which are subject to recall bias and limited precision. The AMES provides a global index of mental energy but may not capture finer attentional processes. Similarly, the PSQI reflects retrospective sleep quality over a month, which may not fully represent immediate sleep states relevant to performance. Incorporating objective tools such as actigraphy for sleep and EEG or eye-tracking for cognitive focus could enhance measurement accuracy and provide clearer insights into the mechanisms linking sleep, AME, and sport performance.

Furthermore, future research could expand its scope to encompass a wider range of sports and incorporate diverse performance metrics to enhance the robustness and generalizability of the findings. Moreover, investigating moderating variables such as gender, competitive stress, psychological resilience, and cultural differences may provide a more nuanced understanding of these phenomena (Juliff et al., 2018). The integration of physiological measures, including hormone assays, metabolic markers, brainwaves and heart rate variability, alongside psychological assessments, could yield comprehensive insights into the complex effects of sleep on athletic performance. Finally, examining the interactions among sleep, AME, and post-competition recovery strategies could provide valuable information for effective athlete management throughout competitive seasons.

Conclusions

The present study provides preliminary evidence for the role of AME as a mediator linking sleep quality with basketball three-point shooting performance and percentage among collegiate basketball players. The results suggest that better sleep quality was modestly associated with improved performance, both directly and indirectly via AME levels. These findings should be interpreted cautiously, as the mediation effects were small in magnitude and based on self-reported measures. Nonetheless, the study highlights AME as a possible psychological pathway through which sleep may contribute to sport-specific performance outcomes. Future research should employ longitudinal design and include a wider range of performance metrics to further clarify the relationship between sleep, AME, and performance, as well as enhance the generalizability of these findings across various athletic populations.

Supplemental Information

Supplemental Information 1 The measures of sleep quality, athletic mental energy and basketball 3-pointer outcomes

Demographic variables, an 18-item Athletic Mental Energy Scale, and a 7-dimensional Sleep Quality assessment. The coding can be identified within the variable names.

Artificial intelligence (AI)-based tools, Grammarly and Wordvice, were employed exclusively to ensure grammatical and spelling accuracy.

Additional Information and Declarations

Competing Interests

Author Contributions

Human Ethics

Data Availability

Frank Jing-Horng Lu is an Academic Editor for PeerJ.

Shu-Yueh Chan conceived and designed the experiments, performed the experiments, prepared figures and/or tables, and approved the final draft.

Wei-Jiun Shen conceived and designed the experiments, performed the experiments, analyzed the data, authored or reviewed drafts of the article, and approved the final draft.

Shin-Liang Lo conceived and designed the experiments, performed the experiments, prepared figures and/or tables, and approved the final draft.

Yun Che Hsieh performed the experiments, analyzed the data, prepared figures and/or tables, and approved the final draft.

Frank J.H. Lu performed the experiments, analyzed the data, authored or reviewed drafts of the article, and approved the final draft.

Garry Kuan analyzed the data, prepared figures and/or tables, authored or reviewed drafts of the article, and approved the final draft.

The following information was supplied relating to ethical approvals (i.e., approving body and any reference numbers):

The Antai-Tian-Sheng Memorial Hospital Institutional Review Board approved the study (TSMHIRB-23-090-B).

The following information was supplied regarding data availability:

The raw data is available in the Supplemental File.

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
