# Peer review of "Effect of quality sleep on basketball three-point shooting outcomes: the mediating role of athletic mental energy in a cross-sectional study of collegiate athletes"

_PeerJ, doi:10.7717/peerj.20355_

## Round 0.1 · original submission · Major Revisions

· Academic Editor

Major Revisions

·

Basic reporting

The manuscript is written in clear, professional, and unambiguous English. The introduction provides sufficient context and clearly defines the scope and relevance of the study. The literature is current, relevant, and appropriately cited. The structure of the manuscript is consistent with PeerJ standards and contributes to overall readability. Figures and tables are high quality, appropriately labeled, and effectively support the text. All raw data are made available in accordance with the journal’s policies. The manuscript is generally well written and clear. However, some issues need attention:

1. There is an inconsistency in terminology regarding participant characteristics. In Table 1, the variable is labeled as "sex", while throughout the manuscript (Lines 217, 295, 298, 299), the term "gender" is used. It is important to distinguish between "sex" (biological) and "gender" (social identity) to maintain conceptual clarity and consistency.
Please revise the terminology to be consistent throughout the manuscript.

2. In Table 2, the abbreviations "PSQI" and "AMES" are used (Line 2), but their full forms are not provided in the table legend or footnote. For clarity and to assist readers unfamiliar with these measures, all abbreviations used in tables should be defined directly beneath the table.
Please add full definitions for all abbreviations used in Table 2 (e.g., PSQI = Pittsburgh Sleep Quality Index; AMES = Athletic Mental Energy Scale) in the table footnote, even if they are defined elsewhere in the text.

Experimental design

The research question is well-defined, original, and addresses a meaningful gap in the literature. The study represents original primary research that falls within the scope of the journal. The methods section is described in sufficient detail to allow for replication, and the study design adheres to the appropriate ethical and scientific standards. The investigation was conducted rigorously, with clear attention to methodological soundness.

Validity of the findings

All analyses appear statistically sound, and results are robust and appropriately interpreted. The underlying data are provided and have been carefully handled. The conclusions are well-aligned with the research question and results, without overstatement. Although novelty is not formally assessed, the study has the potential to make a meaningful contribution to the existing body of knowledge.

Reviewer 2 ·

Basic reporting

• While the article is generally easy to follow, there are sections where grammar should be improved. In particular, the final paragraph of the introduction (Lines 145-156) should be reviewed. I appreciate that English may not be the first language of the authors, and the issues are not to an extent to would prevent publication, but I would recommend checking grammar across the article.
• Introduction: Given that the population used is student-athletes and therefore face challenges to sleep relating to both sport and academic study, it would be beneficial to provide some background into what makes this population different (and makes poor sleep highly prevalent) from elite athletes. There are various articles in the review here that could be used in the introduction (and the discussion) to support this - https://doi.org/10.1007/s40675-025-00341-z
• Introduction: The association with sports performance (Lines 58-62) is highly mixed, with many studies finding no significant effect, as sleep is one of many variables that can influence acute performance. Consider being more critical and providing both perspectives for balance.
• Introduction: Please provide a reference for the quote on Lines 66-67.
• Introduction: Cognitive energetic theory (Line 93) is something that readers may not be familiar with, and more detail is required here. In addition, there needs to be a more explicit link made with AME.
• Introduction: “To enhance the reliability of mediation testing”…(Lines 135-136) – this would be better placed within the methods, or reworded to make a more general point around the use of 3-point shooting in research rather than this specific study.
• Discussion: For the discussion on monitoring technologies, it would be beneficial to consider some of the negative outcomes that are increasingly reported in athletes, such as orthosomnia (e.g., Trabelsi et al., 2023; DOI: 10.1016/j.jshs.2023.02.005).
• Discussion: The final paragraph of the theoretical implications and contributions sub-section (Lines 363-367) largely repeats the preceding paragraphs, and could either be removed or integrated into the conclusion.
• Discussion: The first two paragraphs of the limitations (Lines 406-420) are the same point and could be condensed in size, while the limitation is valid, all study designs come at the expense of not using a different design.
• Discussion: For the limitations, I would consider the questionnaires used. The PSQI asks participants to reflect on sleep over the previous month before performing an acute task – would a questionnaire reporting the previous night (or a self-reported sleep duration the night before) change the results? I am less familiar with the AMES than you will be, but I would expect there to be similar temporal considerations.

Experimental design

• Research aims are clearly outlined, and the experimental design is appropriate to answer the research aims
• Methods: Parts of the ‘Participants’ sub-section (Lines 160-168) could be split more effectively. Lines 160-164 describe sample size estimation and could be a separate sub-section. Parts of lines 167-173 present descriptive data that is shared with results Table 1 and should be consolidated in one location. Lines 169-170 on missing data should be with the remaining data screening in the ‘statistical analysis’ sub-section.
• Methods: The study recruited student-athletes from ‘Division 1 and Division 2 institutions in Taiwan’ (Line 168). Most readers will be unfamiliar with what performance level these will correspond to and could easily be confused with NCAA Div 1 and 2 competitions in the United States. Consider also retrospectively classifying athletes using a framework (e.g., McKay et al., 2022; DOI: 10.1123/ijspp.2021-0451) to help the reader.
• Methods: The details on the demographic questionnaire (Lines 216-218) could be removed or reduced here; they are already presented in the Participants sub-section (or vice-versa and kept here but removed earlier).
• Methods: Lines 227-228. The total PSQI score was utilized for the primary
• 228 analysis, except for the variable concerning sleep medication – Do you mean the PSQI components rather than the global score?
• Methods: From the dataset, it looks like you did your analysis in SPSS; this should be added to the statistical analysis sub-section.

Validity of the findings

• The supporting data has been provided, and the statistical analysis appears to have been appropriately conducted.
• Methods: The data screening approach from Tabachnick + Fidell is suitable for the data (Lines 260-261). More detail should be provided here. Were univariate and multivariate outliers checked? Were any visual inspections of normality conducted?
• Methods: Confidence intervals are presented in Table 3; these need to be added to the methods section.
• Methods: Some details on the experimental setup of the basketball task should be presented in greater detail, such that the experiment could be replicated. What time of day was each trial conducted? Was this before or after a training session, or ad-hoc?
• Methods/Results: All the results related to data screening (Lines 268-274) would be better placed in the statistical analysis section at the end of Line 261, as they are not related to the primary research aims and affect the SEM conducted as discussed in Lines 273-274.

Additional comments

The research study itself appears to have been very well conducted, and to have such a large sample to participate in experimental conditions should be commended.

Reviewer 3 ·

Basic reporting

The authors note as a limitation that "the exclusive focus on basketball 3-point shooting performance may not wholly capture the broader dimensions of sports performance." However, the manuscript would be strengthened if the Introduction provided a more precise explanation of which components of sports performance are reflected in 3-point shooting, along with relevant theoretical justification. Additionally, in the Discussion, it is recommended that the interpretation and implications of the results be logically developed in relation to these performance elements, thereby better contextualizing the significance of the findings.

Experimental design

-

Validity of the findings

While the authors appropriately acknowledge in the Discussion that "the study's cross-sectional design restricts the ability to draw definitive causal conclusions," some statements in the Conclusion and the first paragraph of the "Theoretical implications and contributions" section appear to overstate causality—for example, the phrase "high-quality sleep significantly improves sports performance."
Given the cross-sectional nature of the study, it would be more appropriate to revise such expressions to reflect associations rather than causal effects.

Additional comments

1. It is important to carefully consider the mismatch in the timeframes assessed by each variable. The 3-point shooting test reflects performance on a single day, whereas the PSQI evaluates retrospective sleep quality over the past month. As such, it remains unclear how well the PSQI score reflects the participant's sleep status on the night prior to the shooting test, which may have had a more direct impact on performance (e.g., participants may have had unusually good or poor sleep the night before the test). This discrepancy should be acknowledged as a potential limitation of the study.

2. The scoring method used for the 3-point shooting test in this study appears to be a valid approach for evaluating the quality and gradation of shooting performance in more detail. However, the success rate (i.e., number of successful shots divided by total attempts), which has been widely used in previous research, is arguably more intuitive and accessible to readers, and it aligns more directly with how performance is typically assessed in actual game settings.
Therefore, including a supplementary (sensitivity) analysis using a simple success rate–based performance measure may enhance the interpretability and generalizability of the findings.

3. The presentation of descriptive statistics for items assessed via the demographic questionnaire appears somewhat inconsistent, with some values reported in the "Methods" section and others in the "Results." While reporting age and gender in the "Participants" subsection is appropriate, it would be preferable to present the descriptive statistics for other variables (e.g., years of experience, daily training hours) collectively in the "Results" section for clarity and consistency.
In addition, while the descriptive statistics for the Athletic Mental Energy Scale (AMES) are provided in Table 2, it would be helpful to include a brief comparison with previous studies to clarify whether the observed values fall within expected ranges. This could strengthen the discussion regarding the representativeness of the sample used in the present study.

Reviewer 4 ·

Basic reporting

Title & Abstract
Title: The authors have provided a clear title but have omitted to include the study design. I recommend that this be incorporated for full transparency. A suggested revised title is: “The mediating role of Athletic Mental Energy Scale on quality sleep and basketball 3-point shooting performance: a prospective observational study of college basketball players” or something to that effect.
Abstract, Methods: Page 6, Line 22 (and throughout) – the term ‘mental energy’ is used throughout; however, the authors have actually used the Athletic Mental Energy Scale. This is a specific tool and should therefore be cited and represented as such throughout. Therefore, I recommend that the authors use the term AME scale rather than ‘mental energy’ throughout the paper.
Abstract, Methods: Page 6, Line 26: The methods are incomplete in this section. I recommend that the authors provide further detail on data collection and the principal statistical analysis in this part rather than the opening section of the Results. This would improve the clarity of reporting.
Abstract, Results: Page 6, Line 29-30: The author's interpretation of mental energy having a principal mediating effect is somewhat over-interpreted based on the results and data presented. Whilst there is a signal that this may be the case, this is not particularly strong. Given the depth and breadth of ‘mental energy’, I recommend that the authors consider whether a more targeted mediator is identified, which may offer a stronger interpretation.

Abstract, Conclusion: Page 6, Line 30-32: The authors highlight that sleep quality and sleep hygiene are already acknowledged as important for sport performance. This is correct. This underlines the fact that these results are not particularly new and novel, and therefore, I recommend that the rationale for this study will need to be stronger in the Introduction and Discussion to justify the paper further.

Introduction
Introduction, Page 6, Line 38 to Page 9, Line 156: The Introduction is long and provides too much detail, which would be better placed in the Discussion. I recommend that the authors try to focus the context of the study and rationale to three to four paragraphs. This would increase the focus on the study.
Introduction, Page 8, Line 126 to Page 9, Line 144: This section is crucial for justifying the study. As the authors highlight in the Introduction (to great depth), there is existing knowledge on sleep hygiene and its impact on sports performance. The mediating role of psychology is less clear. The authors should provide a more focused and clearer basis for why sports performance could be influenced by AME and physiological impacts, and how this has not been assessed before. The role of basketball as the vehicle for sports performance is clearly explored, but the justification for AME is less so. I recommend that this be augmented and focused on in this section.

Introduction, Page 9, Line 150: The authors should make the distinction between evidence that builds on and evidence that is new and novel. I recommend that the authors reflect on this difference and offer this within this final objective section, as this would then strengthen the rationale for the study.

Figures & Tables
The figures and tables are clear and legible. These are free from unnecessary modifications.

Experimental design

Material and Methods
Materials and Methods, Page 9, Line 160-163: The sample size parameters are not provided. There is no evidence citations supporting why 68 participants would be sufficient for two parameters or why this was previously used. Providing another citation, i.e., Fritz and MacKinnon (2007), is not sufficient. I recommend that the authors provide the target parameters on which this sample size was made, including variance, as currently, I am unable to replicate this sample size calculation based on the information provided in this section.
Materials and Methods, Page 9, Line 165-168: The authors have not provided sufficient information on the eligibility criteria of their cohort. There are no specific inclusion and exclusion criteria. There is no reference to minimum sleep variance, injury, role/position, height, or anthropometric characteristics. This should be considered in the eligibility criteria and clearly presented in a Table 1 demographics characteristics table in the Results section.
Materials and Methods, Page 10, Line 190: If the authors wished to assess the impact on ‘game time’ performance, it is unclear why they did not control the sleep element and merely assessed game-time performance, i.e., longitudinal assessment of the player performance modifying sleep levels. This should then have answered the direct question the researchers were exploring. This should be considered in the Methods and/or Discussion section.
Materials and Methods, Page 10, Line 195: The testing procedures and apparatus used for this method were appropriate. The use of specific sleep and performance measures was also appropriate and clearly justified and referenced in Materials and Methods, Page 11, Lines 216-250. I have nothing further to recommend in this section.
Materials and Methods, Page 12, Line 252: There is a short section titled ‘experimental control’ – in this study design, I do not think there really is a control group, and therefore I recommend this section be removed for clarity of reporting.
Materials and Methods, Page 12, Line 261: The authors should state how data distribution was determined. What statistical approach was adopted to determine this? I recommend that this be clearly presented in this section to ensure all statistical assumptions were met. Line 270 Kolmogorov-Smirnov test mentioned, but this should be in the Methods section.
Materials and Methods, Page 12, Line 264: I recommend that the authors state what statistical program was used to perform the statistical analyses.

Validity of the findings

Results
Results, Page 12, Line 274-282: The authors have provided between-group differences with Student T-Test results. These are not mentioned in the Methods section. Because the authors have not powered their study for this, I recommend that the authors reflect on whether this is needed to answer the research question and whether the p-values are needed, given that these are potentially underpowered analyses. I recommend that the authors reconsider the presentation of p-values in the text and whether descriptive statistics alone would be more appropriate.
Table 1: There is insufficient information provided on the characterization of the players. I recommend that the authors also provide information on injury profile, height, weight, BMI, and something about general sports performance. If these data are not available, this should be provided as a limitation section in the Discussion.
Results, Page 12, Line 23-291: The authors provide correlation analyses within the second paragraph of the results section. However, there are limited data presented in the text, requiring cross-reference to Table 2. I recommend that the authors provide more data in the text to allow the reader to better interpret these findings.
Results, Page 13, Line 295-299: The investigation of the moderating effect of gender is important, but I recommend that this be placed in the Methods Statistical Analysis section rather than the Results section, to improve the clarity of reporting.
Results, Page 13, Line 302-305: The mediation of sleep quality on sports performance was not particularly strong in this analysis. I recommend the authors reflect on this and consider how confident they are in their conclusion of sleep quality on performance based on AME as the mediation. Based on the current data, I think a more cautious interpretation is needed, and this would radically change the focus of this entire paper.

Discussion
Discussion, Page 13, Line 311-314: I recommend the authors re-explore the size of the mediation analysis reported and consider whether their interpretation is appropriate or whether a more cautious interpretation would be prudent.
Discussion, Page 13, Line 317 to Page 14, Line 367: The authors have used previous literature to underscore their findings. This is appropriate. However, there is an opportunity to explore the robustness of the AME scale in this instance. Given the strength of the mediation may be lesser than the authors suspected, it may be that the AME scale itself was accountable for this, and whether further consideration on the use of other measures of concentration, focus, attention, etc., may be appropriate rather than a more generic ‘mental energy’ measure. I recommend that the authors reflect on this within this section of the Discussion.
Discussion, Page 15, Line 370 to 403: This section should be significantly reduced. Whilst I concur that sleep hygiene is important for sports performance, there is a wealth of previous evidence to support this, the findings from this specific study do not offer insights into what form of sleep improvement should be advocated. There is insufficient new knowledge to justify why a cognitive-behavioral approach to sleep strategies should be adopted (for example). Therefore, I recommend that the authors reduce the size and scope of this section and consider the core practical impact of their findings.
Discussion, Page 16, Line 426-430: I recommend that the final limitation point be removed. This is more related to sleep improvement strategies and not the mediating effect of sleep on performance, i.e., the research question. This removal would improve the focus of this section.

Conclusion
Conclusion, Page 16, Line 433-443: I recommend that the authors revise their Conclusion section. I recommend a more cautious interpretation of the mediating role sleep has on sports performance based on these findings using the AME scale. I recommend that the authors also remove the text related to sleep hygiene strategies and approaches to improve sleep, as that was not the role of this specific study.

---

## Round 0.2 · accepted · Accept

· Academic Editor

Accept

Thank you for your efforts to address the reviewer comments. I am pleased to advise that your manuscript is now suitable for publication.

·

Basic reporting

-

Experimental design

-

Validity of the findings

-

Additional comments

The manuscript is well-written and provides valuable insights into the field. I have no further comments to add.

Reviewer 2 ·

Basic reporting

I am happy that all comments related to basic reporting have been sufficiently addressed; the grammar and flow of the article have been clearly improved.

Experimental design

Strengths and limitations regarding the experimental design have been more explicitly stated in this revision.

Validity of the findings

Comments on the validity of the findings have been addressed to a suitable standard.

Reviewer 3 ·

Basic reporting

-

Experimental design

-

Validity of the findings

-

Reviewer 4 ·

Basic reporting

Title & Abstract
Title: This is clear and an appropriate summary of the research question. The addition of the study design provides further details. I have no additional points to make.
Abstract, Page 7, Line 27-39: The Abstract is clear and provides all essential information required to understand the study design, methods, and analysis. The conclusions drawn are appropriate and reflect the results presented in the analysis. I have nothing further to recommend.

Introduction
Introduction, Page 7, Line 43 to Page 10, Line 161: The Introduction is clear. The basis of the study is clearly outlined with the context offered in relation to previous literature and the current evidence base. The justification for this particular study is adequately communicated. I have nothing further to recommend from this section.

Figures & Tables
The figures and tables were clear and legible. The figures were free from unnecessary modification.

Experimental design

Material and Methods
Methods, Page 10 Line 163 to Page 14, Line 315: The methods undertaken in this study are presented in this section. There is a clear summary of the steps undertaken. There is justification for the sample size and data collection approaches. The methodology is sufficiently robust to answer the research question. The statistical analyses undertaken are appropriate and are able to answer the research question. I have nothing further to recommend from this section.

Validity of the findings

Results
Results, Page 14, Line 317 to Page 17, Line 416: The Results from the analysis answer the planned research question. The use of subheadings in this section is appropriate and well-made. This makes both communicating and interpreting the findings clearer and easier to appreciate. The authors have robustly interpreted the analyses correctly. I have no additional recommendations to this effect.

Discussion
Discussion, Page 17, Lines 418 to Page 22, Lines 613: The Discussion section is clear. The use of literature to compare and contrast the findings is appropriate. The opening summary paragraph is valuable to set the scene for these findings. The authors have used relevant literature to contextualize the findings. Whilst the limitations section is now quite exhaustive and detailed, I believe this is appropriate. I have no additional points to recommend. I feel this Discussion section is appropriate.

Conclusion
Conclusions, Page 23, Lines 615 to 625: This section is appropriate. The Conclusions drawn reflect the findings and are interpreted with suitable caution. I have no additional recommendations.